# Analog Bayesian neural networks are insensitive to the shape of the weight distribution

**Ravi G. Patel**    **T. Patrick Xiao**    **Sapan Agarwal**    **Christopher Bennett**
Sandia National Laboratories
{rgpatel,txiao,sagarwa,cbennet}@sandia.gov

## Abstract

Recent work has demonstrated that Bayesian neural networks (BNN's) trained with mean field variational inference (MFVI) can be implemented in analog hardware, promising orders of magnitude energy savings compared to the standard digital implementations. However, while Gaussians are typically used as the variational distribution in MFVI, it is difficult to precisely control the shape of the noise distributions produced by sampling analog devices. This paper introduces a method for MFVI training using real device noise as the variational distribution. Furthermore, we demonstrate empirically that the predictive distributions from BNN's with the same weight means and variances converge to the same distribution, regardless of the shape of the variational distribution. This result suggests that analog device designers do not need to consider the shape of the device noise distribution when hardware-implementing BNNs performing MFVI.

## 1 Introduction

Uncertainty quantification is an essential capability for machine learning (ML) algorithms, particularly where these models are in the critical path of high-consequence or safety-critical decisions [1]. Without being able to accurately assess uncertainty, ML models cannot be trusted and therefore cannot be deployed in such systems. The Bayesian neural network (BNN) possesses the generalization capability of deep neural networks (DNNs), while providing rigorous estimates of predictive uncertainty by encoding its weights as trainable probability distributions rather than as fixed parameters [2, 3]. BNNs can capture uncertainty arising from ambiguous data, as well as uncertainty that comes from making inferences on data that lies well outside the distribution represented by the training data. Training of BNNs can be made computationally tractable by using variational inference [4, 5], a technique which constrains BNN weights to parameters with only a few tunable values, rather than arbitrarily complex distributions. Nonetheless, large BNN networks are difficult to implement at scale due to their complexity and sampling requirements; for every prediction from a BNN, a large number of random numbers must be sampled – at least one for every weight – and these random number generations can be prohibitively slow and power-intensive for these applications [6]. Unfortunately, there are many target systems where uncertainty quantification is critical to enable safe autonomy, yet where ML algorithms must be processed in real time and within a small power envelope (typically less than 1W). Given this constraint, these methods can scarcely be deployed in modern edge or mobile devices.

To address this challenge with BNN inference on edge systems, a promising approach is to map BNNs onto energy-efficient analog in-memory computing (AIMC) hardware. This computing paradigm encodes weights as individual conductances within a larger crossbar array of programmable memory devices, and uses analog circuit laws to physically compute matrix-vector multiplications (MVMs) [7]. Though AIMC has so far been applied only to conventional deterministic DNNs, early proposals have suggested that designers could exploit the noise inherent to emerging

Second Workshop on Machine Learning with New Compute Paradigms at NeurIPS 2024 (MLNCP 2024).

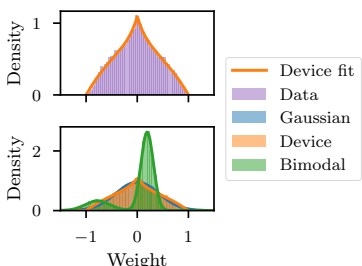

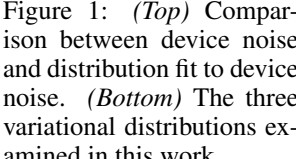

Figure 1: *(Top)* Comparison between device noise and distribution fit to device noise. *(Bottom)* The three variational distributions examined in this work.

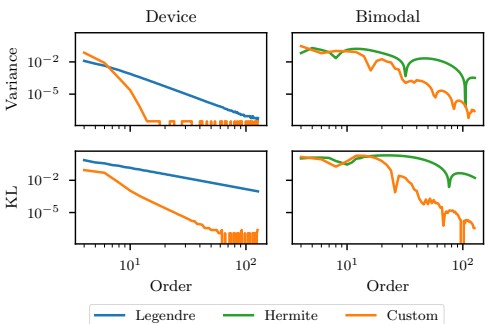

Figure 2: Squared difference of order approximation and previous order approximation. Custom quadratures *(Orange)* converge faster than standard quadratures *(blue)* in estimating variance *(Top)* and KL divergence to a Gaussian *(Bottom)*.

memory devices to implement probabilistic algorithms [8]. In particular, the conductances of memory devices have some level of intrinsic analog noise, which provides the opportunity to sample many analog random numbers and perform analog matrix computations on these random numbers in parallel [9]. This approach is restricted to mean field variational inference

Critically, implementing BNNs on stochastic AIMC hardware requires the distribution of conductance noise in each individual memory device to be tunable (writable) to reflect the full range of probability distributions obtained from the software BNN training. Yet the noise distributions of real memory devices are often non-ideal; they may be minimally tunable, having only one or two degrees of freedom, or sampling them may yield an exotic distribution that isn't like the Gaussian distribution commonly used in VI. Prior work has nonetheless trained BNNs with Gaussian distributions and approximated them during inference time using non-ideal memory device device noise distributions [9, 10, 11, 12, 13]. Although this induces a significant amount of error at the level of a single probability distribution, the UQ end-application metrics from BNNs built from these non-ideal sampling devices (*e.g.*, expected calibration error) has been observed to be remarkably resilient to these errors.

In this work, we present a method to train BNNs using variational inference that accounts for the non-standard probability distributions of actual noisy analog memory devices. This method effectively eliminates the approximation error involved in transferring the trained probability distributions in a BNN to the available conductance noise distributions in analog hardware. Nonetheless, we use this method to show numerically that the approximation error does not have a significant impact on the predictive uncertainty of deep BNNs, due to the effect of the central limit theorem (CLT) within each layer. This result provides a theoretical basis for the conclusion that large arrays of stochastic memory devices that do not have a high degree of tunability or precision in their encoded probability distributions can nonetheless be used to efficiently process large-scale BNNs.

## 2   Predictive distributions using device weights

A crossbar of memory devices with tunable mean conductances ($\mu$) and tunable conductance noise ($\sigma$) enables the efficient computation of a stochastic MVM. In this case, the multiplication of a vector occurs with a matrix sampling from known, unique probability distributions. Often, the origin of randomness in the analog system is driven by thermal fluctuation in the conductance value of a given memory device; *e.g.*, noise can can appear as complex $1/f$ noise fingerprints in filamentary resistive random access memory (ReRAM) [14] . Using a novel bit-cell design for stochastic MVMs introduced in [9], the mean conductance and the conductance noise do not need to be independently controllable within a single device. Stochastic MVMs have been demonstrated or simulated using magnetic random access memory (MRAM) [9], resistive random access memory (ReRAM) [13, 15, 10], charge-trap memory [11], and electrochemical random access memory (ECRAM) devices [12] that were engineered to enable operation over a range of programmable noise values ($\sigma$). Notably, among all these proposed devices, the shape of the noise probability distribution is not tunable in general, but rather a function of the underlying device physics of that memory component. Following

a number of individual sampling opreations, the analog result of a stochastic MVM is converted from raw current and current noise into a vector of digital output signals using companion analog-to-digital converters (ADCs) at the bottom of each column in the array.

For the remaining results in this paper, we use the noise probability distribution of the Bayes-magnetic tunnel junction (Bayes-MTJ) device from [9]. This device exploits the fact that the magnetization of a circular in-plane MTJ has two easy axes and no energetically preferred orientation. Therefore, the magnetization can rotate randomly due to thermal fluctuations, resulting in random changes in the tunnel magnetoresistance (conductance) of the MTJ. This noise has a distribution that is not perfectly described by a standard Gaussian distribution, as in Fig. 1. By modulating the built-in voltage inside the Bayes-MTJ, the width of the distribution could be made large or small without affecting the distribution's shape. Crossbar arrays of Bayes-MTJs, combined with crossbar arrays of less noisy multi-state MRAM devices such as domain-wall MTJs, can be used together to implement stochastic MVMs and perform BNN inference.

In addition to the real device noise, we also examine inference with a weight distribution much further from a Gaussian than device distribution, a mixture of Gaussians. We refer to this distribution as the bimodal distribution. Therefore as in Figure 1 bottom pane, we contrast these two designer distributions against a standard Gaussian distribution throughout the remainder of this work.

## 3 Mean field variational inference with device distributions

Our goal is to compare inference using Gaussians as the variational distribution to realistic device noise as the variational distribution. In VI, one minimizes the Kullback-Leibler (KL) divergence [5], $\min_\alpha KL(p_V(\theta; \alpha) || p_{post}(\theta | X))$, between a variational distribution, $p_V$, and the posterior, $p_{post}(\theta | X) \propto p_\ell(X|\theta)p_{prior}(\theta)$ where $X$ is data, $\theta$ is a vector of model parameters, $\alpha$ is the variational parameters, $p_\ell$ is a likelihood, and $p_{prior}$ is a prior, by maximizing the evidence lower bound (ELBO),

$$\min_\alpha \text{ELBO}(\alpha) = E_{p_V}[\log p_\ell(X|\theta)] - D_{KL}(p_V || p_{prior}), \tag{1}$$

The expectation of first term in (1) is high dimensional but can be estimated by Monte Carlo (MC), $E_{p_V}[\log p_\ell(X|\theta)] \approx \frac{1}{N}\sum_i^N \log p_\ell(X|\theta_i)$, sampling the parameters from the variational distribution, $\theta_i \sim p_V$. We will work with the mean field assumption, i.e. assume the parameters are all independently distributed. Furthermore, we will assume each parameter is distributed by a scaled and shifted version of the same base distribution, $q_V$. This allows us to use the reparameterization trick to compute the MC estimate. We sample, $z_{ij} \sim q_V$, per parameter, per Monte Carlo sample, and transform each vector $z_i$ as $\theta_i = \sigma z_i + \mu$ where and $\mu$ and $\sigma$ are vectors of shift and scale variational parameters. Section 3.1.3 discusses our approach to sampling $q_V$ for the device distribution.

The mean field assumption and reparameterization trick also allow us to simplify the KL divergence between the variational distribution and the prior in (1). We also choose an i.i.d. prior, $p_{prior}(\theta) = \prod_i \hat{p}(\theta_i)$. Due to the mean field approximation, the integrals in the second term of (1) are one dimensional. There are closed form expressions for Gaussian variational distributions and Gaussian priors, but we must use quadrature for the device distribution. See Section 3.1.2 for further details. The variational inference optimization problem becomes,

$$\min_{\sigma_j, \mu_j} \frac{1}{N}\sum_i^N \log p_\ell(X|\sigma_j z_{ij} + \mu_j) - \sum_j \left( \log \sigma_j + \int q(z) \log \hat{p}_{prior}(\sigma_j z + \mu_j)dz \right) \tag{2}$$

where $z_{ij} \sim q$. We have drop terms constant in $\mu$ and $\sigma$ and simplified.

### 3.1 Numerical Approximations

This section details various numerical approximations used throughout this work. The device distribution needs careful numerical analysis to generate samples from and compute expectations with.

### 3.1.1 Maximum likelihood fit to device noise

We compare the predictive distributions using three different neural network weight distributions, Gaussian, device, and a mixture of two Gaussians (bimodal). We model the device noise distribution

as,

$$q_D(x) = A\exp\left(\frac{-|x|}{B}\right) - A\exp\left(\frac{-1}{B}\right) + C(1-x^2) \qquad -1 \le x \le 1$$
$$q_D(x) = 0 \qquad\qquad\qquad\qquad\qquad \text{else} \tag{3}$$

and find the maximum likelihood estimates of $A, B$, and $C$ using the Adam optimizer [16] under the constraint that $\int_{-1}^{1} q_D(x)dx = 1$. Throughout this work, we compare results using this device distribution to a Gaussian distribution, $q_G$, and a mixture of Gaussians, $q_M$, with the same mean and standard deviations. Figure 1 demonstrates the device distribution fit and compares the three variational distributions.

### 3.1.2 Quadrature

In general, there is no closed form expression for expectations with respect to the device distribution,

$$E_q[f(x)] = \int_{-1}^{1} f(x)q(x)dx \tag{4}$$

so we need to use approximations. Of particular importance is approximating variance, $E[(x - E[x])^2]$, and the KL divergence from variational distributions to other distributions, $D_{KL}(q||p) = E_q[\log q - \log p]$.

We can efficiently and accurately integrate expectations of polynomials and smooth functions using the device distribution as the weighting function in Gaussian quadrature. In fact, cross entropies with respect to the Gaussian distribution can be exactly integrated with only two quadrature points. In other cases, we design custom quadrature rules based on standard quadratures. Figure 2 demonstrates the efficacy of these quadratures. More details on this formulation and justification for this choice are given in Appendix B.

### 3.1.3 Inverse sampling

We generate additional device noise samples using inverse transform sampling. After fitting to device noise, we have access to the CDF, $Q_D(x) = \int_{-1}^{x} q_D(z)dz$. We can generate new samples by applying the inverse CDF, $G = Q_D^{-1}$, to samples from the uniform distribution,

$$u \sim U[0,1] \Rightarrow x = G(u) \Rightarrow x \sim q_D \tag{5}$$

We cannot find a closed-form expression for the inverse CDF and must use numerical approximation. Because $q_D$ is symmetric across $x = 0$, $Q_D$ and $G$ are symmetric about 180 degree rotations around $(x, u) = (0, \frac{1}{2})$. Therefore, we can approximate a restriction of the inverse CDF, $\hat{G} : [-1, 0] \rightarrow [0, \frac{1}{2}]$, and reuse it for the other halves of the full domain and codomain.

$\hat{G}$ is smooth and amenable to polynomial approximation in the domain $[\epsilon, \frac{1}{2}]$ for $0 < \epsilon \ll \frac{1}{2}$. However, near $u = 0$ the function is not differentiable. By differentiating $Q_D(\hat{G}(u)) = u$, we find,

$$\lim_{u\to 0^+} Q_D(\hat{G}(u))' = \lim_{u\to 0^+} Q_D'|_{\hat{G}(u)} \left.\hat{G}'\right|_u = 1 \Rightarrow \lim_{u\to 0^+} \left.\hat{G}'\right|_u = \lim_{x\to -1} \frac{1}{q_D(x)} = \lim_{x\to -1} \frac{1}{C(1-x^2)} \tag{6}$$

This limit diverges due to the quadratic term. Instead of approximating $\hat{G}$ directly, we use polynomials to approximate a function with the non-differentiability subtracted and then add the non-differentiability back in.

The CDF has the 2nd order Taylor expansion near $x = -1$, $Q_2(-1 + x) = Q_D(-1) + Q_D'|_{-1} x + \frac{1}{2} Q_D''|_{-1} x^2$, which can be inverted for $x \in [-1, 0]$ to give, $\hat{G}_2 = Q_2^{-1}$. See Appendix A for the formal expressions of $Q_2$ and $\hat{G}_2$. We fit a Legendre polynomial, shifted and scaled to the restricted domain, to the difference, $\tilde{G} = \hat{G} - \hat{G}_2 \approx \sum_i c_i \phi_i$, at the Gauss-Lobatto points. We employ Brent's root-finding method [17] to compute $\tilde{G}$ at these points. The restriction of the inverse CDF is then approximated as the sum, $\hat{G} = \sum_i c_i \phi_i + \hat{G}_2$. To approximate the restriction of $G$ to $\cdot : [\frac{1}{2}, 1] \rightarrow [0, 1]$, we rotate $\hat{G}$ 180 degrees around $(x, u) = (0, \frac{1}{2})$. The full approximation of $G$ is,

$$G(u) = \begin{cases} \sum_i c_i \phi_i(u) + \hat{G}_2(u) & u < \frac{1}{2} \\ -\sum_i c_i \phi_i(1-u) - \hat{G}_2(1-u) & u > \frac{1}{2} \\ 0 & u = \frac{1}{2} \end{cases} \tag{7}$$

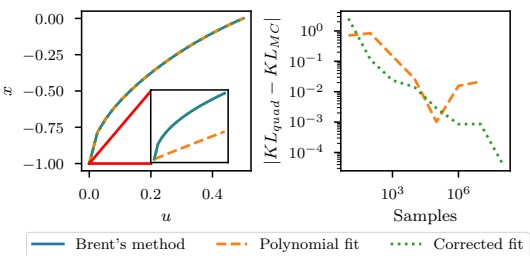

Figure 3: Comparison between corrected fit and polynomial fits to inverse CDF. *(Left)* The approximations to inverse CDF over the full domain. *(Center)* The approximations near $u = 0$. *(Right)* Absolute difference between Monte Carlo and quadrature estimates for the KL divergence between the device distribution and a centered Gaussian with the same variance. The corrected fit performs better than the polynomial fit.

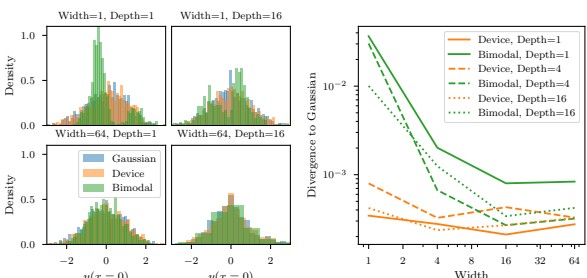

Figure 4: *(Left)* Comparison of three predictive distributions for energy distance problem at different widths and depths. *(Right)* KL divergence from predictive distributions for $y(x = 0)$ using Device and Bimodal weights to predictive distributions from Gaussian weights.

The left subplot in Figure 3 show that the corrected fit performs better than the direct fit. We also compare them by evaluating Monte Carlo estimates of the KL divergence from the device distribution to a Gaussian of the same mean and variance. While Monte Carlo estimates using the polynomial fit to the inverse CDF appears to converge to a biased estimate for the KL divergence while the correct fit continues to converge to the quadrature approximation of the KL divergence.

## 4 Assessing the impact of diverse variational distributions on neural network performance

In this section we compare the predictive distributions from probabilistic neural networks using the three variational distributions. See Appendix C for details on training and the architectures.

### 4.1 Energy distance minimization

Firstly, we trained a dense neural network function from reals to reals, $f_{nn} : x; \theta \mapsto y$, with Gaussian weights, $\theta^G$, to match the normal distribution at $x = 0$. This set-up allows us to control the form of the predictive distribution and evaluate inference when we switch between various variational distributions. We minimize the energy distance with respect to the variational parameters,

$$\min_{\mu,\sigma} 2E_{\substack{\theta_i^G \sim p_G \\ y_j \sim p_T}} \left[ |f_{nn}(0; \theta_i^G) - y_j|^\alpha \right] - E_{\theta_i^G, \theta_j^G \sim p_G} \left[ |f_{nn}(0; \theta_i^G) - f_{nn}(0; \theta_j^G)|^\alpha \right] \qquad (8)$$

with $\alpha = 1.5$. We use MC to estimate expectations over the neural network outputs and quadrature with the target distribution as the weighting function for expectations over the target distribution. After training, we compare predictive distributions at $x = 0$ for the same network and variational parameters, replacing the base distribution from Gaussian to either device or bimodal. In Figure 4, we vary the neural network depth and width. We observe that the predictive distributions for the three distributions converge to the target distribution as the width increases.

### 4.2 Scalar Regression

Next, we performed Bayesian 1D regression using the variational inference framework described above. We choose as our true model,

$$y = \sin(2\pi x) + \epsilon \qquad \epsilon \sim N(0, 0.1(1 + \min(0, x - 1))) \qquad (9)$$

We generate training data, $x_{train} \sim u[-1, 1]$, and compute $y_{train}$ from the above model. Likewise, we compute test data using, $x_{test} \sim u[-1.5, 1.5]$ and $y_{test}$ from the same model. We next fit the data with MFVI training using Gaussians as our variational distribution and compare the posterior

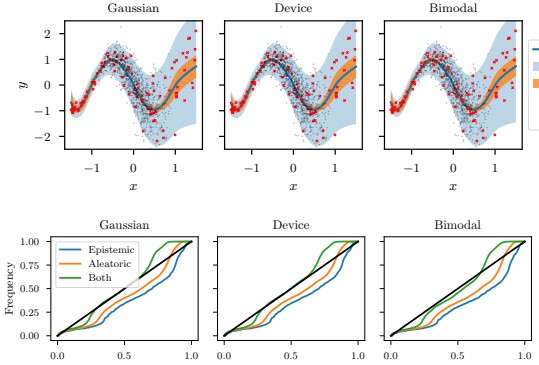

Figure 5: The predictive distributions from all three weight distributions are nearly identical for the scalar regression problem. $\sigma_a$ is the aleatoric error predicted by $f'_{nn}$ and $\sigma_e$ is the empirical standard deviation of the mean predictions under resampled neural network weights, $\xi$.

Figure 6: Calibration curves for three distributions during inference on the test data after training with the Gaussian distribution. The RMSE error for all cases on the test data is around 8%.

predictive distributions obtained by replace the Gaussians with device and bimodal distributions of the same means and variances.

We choose as our predictive model a feed-forward, Bayesian neural network. We model the error as heteroscadastic noise predicted by a feed-forward, deterministic neural network,

$$\xi \sim p_V \qquad y_{pred} - y \sim N(0, \sigma_a) \qquad y_{pred} = f_{nn}(x; \xi) \qquad \sigma_a = f'_{nn}(x, \xi') \qquad (10)$$

The model is similar to [18], except that the aleatoric predictions are deterministic. After training, in Figure 5 we compare inference using the three variational distributions by fixing the variational parameters, but replacing the form of the Gaussian base distribution to the device distribution and the bimodal distribution. At inference, the predictive distributions, with this change is nearly identical to the predictive distribution produced by the Gaussians.

### 4.3 UTKFACE

Lastly, we performed Bayesian inference on the UTKFACE dataset [19]. We seek a mapping from images of faces to ages and use the same model as (10), replacing the feedforward neural networks with convolutional neural networks. As in the last example, we train with the Gaussian distribution, but replace the variational distribution with the device and bimodal distributions. Figure6 compares the calibration curves obtained from inference on test data. They are obtained as discussed in [18]. We find that the predictions obtained by any of the variational distributions used during inference are identical, regardless of the distribution used during training.

## 5 Discussion and Conclusions

We empirically demonstrate that the shape of the variational distribution has little impact on the posterior predictive distributions in mean field variational inference; only the mean and variance is important. As it is difficult to tune the shape of the variational distribution in hardware, these results suggest that device engineering efforts do not need to focus on producing specific sampling curves for downstream variational distributions. Furthermore, one may use the standard Gaussian variational distribution to train the networks in software and reuse the scale and shift parameters in hardware, regardless of the shape of the hardware noise distribution. Our work enables this by establishing methods for transforming predictive distributions. While, as was shown in Fig 4., sufficiently wide neural network layers are still necessary to reduce the difference between predictive distributions, the required width of networks to do so is minimal due to the CLT. Future work will examine whether any loss exists in narrow, deep networks on tasks of interest. Additionally, our claims could be verified in hardware within small probabilistic arrays.

## Acknowledgments

We use TensorFlow [20] for training neural networks, NumPy [21], SciPy [22], and Chaospy [23] for special functions and polynomial approximation, SymPy [24] for symbolic calculations, and GNU Parallel to run experiments in parallel [25].

Sandia National Laboratories is a multi-mission laboratory managed and operated by National Technology and Engineering Solutions of Sandia, LLC., a wholly owned subsidiary of Honeywell International, Inc., for the U.S. Department of Energy's National Nuclear Security Administration under contract DE-NA0003525. This paper describes objective technical results and analysis. Any subjective views or opinions that might be expressed in the paper do not necessarily represent the views of the U.S. Department of Energy or the United States Government.

SAND number: SAND2024-16263C

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

## A  Formula for $P_2$ and $\hat{G}_2$

In Section 3.1.3 we utilize the second order Taylor expansion of the device CDF,

$$P_2(-1+x) = \frac{\left(A + 2BCe^{\frac{1}{B}}\right)(x+1)^2 e^{-\frac{1}{B}}}{2B} \tag{11}$$

which has inverse for $x \in [-1, 0]$,

$$\hat{G}_2(0+u) = \frac{-A - 2BCe^{\frac{1}{B}} + \sqrt{2}\sqrt{Bu\left(A + 2BCe^{\frac{1}{B}}\right)e^{\frac{1}{B}}}}{A + 2BCe^{\frac{1}{B}}} \tag{12}$$

to approximate the inverse CDF.

## B  Further details on quadrature

If we use the device distribution as the weighting function in (4), quadrature rule of $N = 2$ abscissas and weights is sufficient to exactly integrate polynomials up to order $2N - 1 = 3$. Since the log likelihood for a Gaussian is a quadratic polynomial we can integrate the cross entropy, $H(q, N) = -E_q[\log N]$, where $N$ is a Gaussian to machine precision. We use this quadrature for variational training with the device distribution since $H(q, N)$ is the only integral needed for training. We use Wheeler's algorithm [26, 27] combined with symbolic calculations for the moments of the device distribution to compute the abscissas and weights for the quadrature. We note that the custom quadratures below perform better for expectations over non-smooth quantities, particularly ones with singularities, e.g., the KL divergence. Additionally, Wheeler's algorithm is numerically unstable for large quadratures, $N \approx 10$. We resort to the quadratures below as needed.

$q_D$ is smooth, so expectations and amenable to Gaussian quadrature everywhere except near $x = 0$ where it is not differentiable. Therefore, we use Gauss-Legendre quadratures in $[-1, -\epsilon]$ and $[\epsilon, 1]$ and the trapezoidal rule in $[-\epsilon, \epsilon]$. While we might expect the Gauss-Hermite quadrature to perform well for $q_M$ because it is $C^\infty$ on $\mathbb{R}$, we found that a two sided approach, summing the Gauss-Laguerre quadratures of $E_{q_D}[f(x)]$ and its reflection across $x = 0$ on $[0, \infty]$, to perform better.

## C  Details on training and architectures

We use the Adam optimizer with TensorFlow's default hyper-parameters for all optimization problems.

For Section 4.1, we use the ELU activation function [28] and train for 10000 iterations with a batch size of 100.

We again use the ELU activation function for Section 4.2 and a dense neural network with 4 hidden layers of width 16. We initialize the network to the maximum likelihood estimate (MLE) using deterministic training on the mean square error. We next fix the biases to the MLE and train the kernels with VI. We train on 10000 data points for 10 epochs with a batch size of 100. We additionally add tempering on the likelihood to allow the optimizer to better explore the variational parameter space, reducing the temperature as $T = \exp(-i/1000)$, where $i$ is the iteration number.

The training method for Section 4.3 is identical to Section 4.2, except we use the ReLU activation function and a convolutional neural network architecture. We use 4 sets of convolution blocks consisting of,

$$\text{Conv} \rightarrow \text{MaxPool} \rightarrow \text{BatchNorm} \rightarrow \text{Conv} \rightarrow \text{MaxPool} \rightarrow \text{BatchNorm}$$

where first convolution in each block increases the filter size by a factor of two. All convolutions use a kernel size of 3 and pools use windows of size 2. These convolution blocks are followed by a reshape to a vector and three dense blocks,

$$\text{Dense} \rightarrow \text{BatchNorm}$$

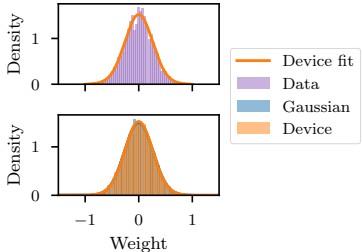

Figure 7: *(Top)* Comparison between device noise and distribution fit to device noise. *(Bottom)* A Gaussian variational distribution and the device distribution.

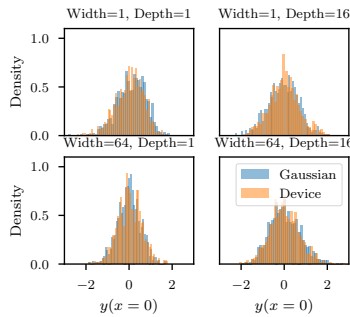

Figure 8: Comparison of predictive distributions from Gaussian and device variational distributions for energy distance problem at different widths and depths.

where the first maps the output of the convolution blocks to dimension 100, the second to dimension 10, and the third to dimension 1. Finally, the output is scaled to have range $[0, 1]$ with a sigmoid function.

## D   Results using tunable-noise ECRAM device

While the bulk of our work focuses on the Bayes-MTJ device, we also find that the predictive distributions produced by a second tunable noise device also converge to the same predictive distribution produced by Gaussian weights. This second device is a vanadium oxide ECRAM device that exploits the material's metal-to-insulator transition to access a wide programmable range of both the conductance and the conductance noise variance, as described by Oh *et al.* [12]. The shape of the noise distribution is qualitatively different from that of the Bayes-MTJ device. Figure 7 shows the results of a MLE fit to the ECRAM device noise using the following parameterization,

$$q_D(x) = A \exp\left(\frac{-x^2}{B}\right) - A \exp\left(\frac{-1}{B}\right) + C(1 - x^2) \qquad -1 \le x \le 1$$
$$q_D(x) = 0 \qquad\qquad\qquad \text{else} \tag{13}$$

Figure 8 repeats the energy score study discussed in Section 4.1 for this device. Note that the device distribution and Gaussian are already similar and the predictive distributions match for even shallow, narrow neural networks.

