# OpenReview forum: "Analog Bayesian neural networks are insensitive to the shape of the weight distribution"
_NeurIPS.cc/2024/Workshop/MLNCP — MLNCP Poster_

### Official Review · Reviewer_Eqw6 · 2024-10-02
**Good work on Analog BNNs**

**Rating:** 8
**Confidence:** 3

**Review:**

The authors discuss Bayesian Neural Networks (BNNs) and explore the potentials of implementing them in analog hardware. Specifically, they consider the problem of controlling the shape of weight distributions in such implementations. The key issue they consider is that traditional BNNs often use Gaussian distributions for their weights, but in analog hardware, the device noise can result in distributions that are not easily modeled by standard Gaussians.

The paper is well-thought and well organized however, there are several typos.

Strong points:

1. The paper maintains a good balance between theoretical and practical insights, which adds credibility to the empirical findings.
2. The claim that the shape of the weight distribution does not matter as long as the first two moments (mean and variance) match is an interesting insight.

Weak points:

1.  The paper could benefit from studying alternative variational inference methods or comparing how other uncertainty quantification methods might behave under different noise models. Would methods like Monte Carlo Dropout or Bayes by Backprop also be insensitive to weight distribution shape?

2. The authors seem to compare the device noise only with Gaussian and bimodal distribution. The paper might benefit from a discussion on other potentially relevant noise distributions, particularly those that could arise in real-world analog hardware implementations.

---

### Official Review · Reviewer_BUUX · 2024-10-04
**Well motivated work on Bayesian Neural Networks in Analog Hardware**

**Rating:** 7
**Confidence:** 3

**Review:**

Summary:
This paper explores an interesting topic on the implementation of Bayesian Neural Networks (BNNs) on analog hardware with a focus on mean-field variational inference (MFVI). The authors claim that the shape of the weight distribution in analog devices does not significantly affect the predictive distribution when the mean and variance are preserved. This insight could greatly influence the design of energy-efficient AI systems.

Strength:
-The authors showed empirical results that are well supported by simulations and experiments on hardware noise that highlight the robustness of BNNs to variational distribution shape variations. The consistent convergence of predictive distributions with varying noise characteristics is a valuable and insightful finding.
-From a theoretical perspective, the use of the Central Limit Theorem (CLT) to explain why the predictive distributions converge is solid and provides a strong well-grounded explanation for the practical results.

Limitations and Questions:
- It would be great if they could show thorough validation on actual hardware platforms. Theoretical predictions and simulations are useful but might not fully capture the complexities of real-world device implementations.
- Would other possible inference methods or hardware platforms benefit from the same insights? While the paper dives deep into the mathematical framework of MFVI, some explanations could benefit from simplification.
- To address some potential scalability issues in terms of network size or hardware complexity for larger BNNs could be a great addition for future direction.

---

### Decision · Program_Chairs · 2024-10-10

Accept (Poster)